# Titanium-Based Microbolometers: Control of Spatial Profile of Terahertz Emission in Weak Power Sources

**Linas Minkevičius [1],\* , Liang Qi [1] , Agnieszka Siemion [2] , Domas Jokubauskis [1] , Aleksander Sešek [3] , Andrej Švigelj [3], Janez Trontelj [3], Dalius Seliuta [1], Irmantas Kašalynas [1] and Gintaras Valušis [1]**

1 Department of Optoelectronics, Center for Physical Sciences and Technology, Saulėtekio Ave. 3, 10257 Vilnius, Lithuania; liang.qi@ftmc.lt (L.Q.); domas.jokubauskis@ftmc.lt (D.J.); dalius.seliuta@ftmc.lt (D.S.); irmantas.kasalynas@ftmc.lt (I.K.); gintaras.valusis@ftmc.lt (G.V.)
2 Faculty of Physics, Warsaw University of Technology, 75 Koszykowa, 00662 Warsaw, Poland; agnieszka.siemion@pw.edu.pl
3 Laboratory for Microelectronics, Faculty of Electrical Engineering, University of Ljubljana, Tržaška 25, 1000 Ljubljana, Slovenia; aleksander.sesek@fe.uni-lj.si (A.S.); andrej.svigelj@fe.uni-lj.si (A.Š.); janez.trontelj1@guest.arnes.si (J.T.)
\* Correspondence: linas.minkevicius@ftmc.lt; Tel.: +370-5-243-1200

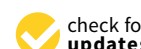

**Featured Application:** **Convenient and easy-to-use detectors for accurate adjustment and control of spatial mode profiles without additional focusing components for various weak power THz sources operating either in time-domain or continuous wave regimes.**

**Abstract:** Terahertz (THz) imaging and spectroscopy set-ups require fine optical alignment or precise control of spatial mode profile. We demonstrate universal, convenient and easy-to-use imaging—resonant and broadband antenna coupled ultrasensitive titanium-based—dedicated to accurately adjust and control spatial mode profiles without additional focusing optical components of weak power THz sources. Versatile operation of the devices is shown using different kinds of THz—electronic multiplier sources, optical THz mixer-based frequency domain and femtosecond optoelectronic THz time-domain spectrometers as well as optically pumped molecular THz laser. Features of the microbolometers within 0.15–0.6 THz range are exposed and discussed, their ability to detect spatial mode profiles beyond the antennas resonances, up to 2.52 THz, are explored. Polarization-sensitive mode control possibilities are examined in details. The suitability of the resonant antenna-coupled microbolometers to resolve low-absorbing objects at 0.3 THz is revealed via direct, dark field and phase contrast imaging techniques as well.

**Keywords:** microbolometer; terahertz sensing; terahertz imaging systems; dark field method; phase contrast method

## 1. Introduction

Terahertz (THz) imaging and spectroscopy revealed itself as promising technologies for a large variety of applications [1]. Because of non-ionizing nature of THz radiation, imaging in this range can be found as a beneficial tool for medicine and biomedical stuff inspection [2–4]. As THz frequencies display the ability to propagate in dielectric medium without strong attenuation, THz imaging and spectroscopy can also serve as valuable instruments for security needs [5–7], post packages inspection [8–10] or industrial materials control and their identification [11–14]. However, direct implementation of THz technology still meets severe difficulties related mainly to reliability of active components (emitters and detectors) and issues of precise alignment of the optical system due to the

presence, as a rule, of passive bulky optical elements like mirrors, lenses or beam splitters. Thus, effective and convenient operation under real environmental conditions still needs to overcome alignment and compact optics problems [15]. On the other hand, it is of decisive demand to provide adequate and favourable control of the spatial mode profile of illumination. In particular, it has strong impact in THz spectroscopy for quality of recorded THz spectra in increasing its bandwidth, lowering the noise floor, as well as by reducing uncertainties in observational data [16–18]. It displays also important role in THz imaging, especially, in examining bulky objects when extraordinary means are needed to be applied in THz beam engineering aiming to provide appropriate image quality [19,20].

Spatial mode profiles of discrete THz spectrum radiation from backward wave oscillators and quantum cascade lasers were already studied by employing thin film vanadium oxide (VOx) based bolometers [21]. Moreover, image of the beam emitted by an optoelectronical THz time-domain spectrometer was recorded using a camera integrating an array of THz optimized antenna-coupled amorphous silicon-based microbolometers [22]. Indeed, it is of distinct interest as THz time-domain spectrometers become widespread, they require precise alignment to get well collimated THz beams, hence, it is beneficial to engage easy-to-use imaging detector to accurately adjust these optical systems. The sensitivity of the presented microbolometers allowed to register images of spatial mode profiles as well as video images despite the fact that their average power was in the range of a few μW [22].

The developed and described here antenna-coupled titanium-based microbolometer (Ti-mB) [23] was initially applied in compact THz imaging systems for security aims [24,25]. Its high sensitivity was successfully exploited also in medical applications via direct THz imaging of carcinoma affected tissues [26].

In a given work, by coupling of the titanium microbolometers with relevant resonant antennas, we modified these sensitive detectors into universal, convenient and easy-to-use imaging instruments well-suited to accurately adjust weak power THz sources without additional focusing components. They can be employed in imaging systems operating either in time-domain or in continuous wave modes for precise alignment and control of their spatial mode profiles. Versatile operation of the devices is displayed engaging different kinds of THz emitters – electronic multiplier sources (THz ELS), optical THz mixer-based frequency domain (THz FDS) and femtosecond optoelectronic THz time domain (THz TDS) spectrometers as well as optically pumped molecular THz laser (THz OPML). Features of the microbolometers coupled with resonant antennas in subTHz range, within the 0.15–0.6 THz range, are exposed and discussed here. The ability to monitor spatial mode profile even far above the antennas resonances, up to 2.52 THz, is explored. Moreover, the possibilities of polarization-sensitive mode control are examined in details, and the recorded mode control results are compared with that obtained using commercially available pyroelectric sensors. In addition, the suitability of the resonant antenna coupled Ti-mB to resolve low-absorbing objects at 0.3 THz is revealed via direct, dark field and phase contrast imaging techniques.

## 2. Bolometers' Design and Experimental Set-Ups

The design of antenna-coupled Ti-mB and its cross section are presented in Figure 1. As it can be seen, the device is constructed using air-bridged approach. The dual dipole THz antenna is coupled to the bolometer and serves as a filter that aims to efficiently collect the incident THz radiation and convert its energy to temperature change. The Ti-mB and THz antenna were processed on a few microns thin SiN membrane to enhance the sensitivity and the response time via reduction of thermal losses of the instrument. The metalized bottom plate under the antenna-bolometer acts as reflecting mirror, additionally amplifying the received signal on the dipole. More detailed description on the design and the technology used to manufacture such devices can be found in Refs. [23,24].

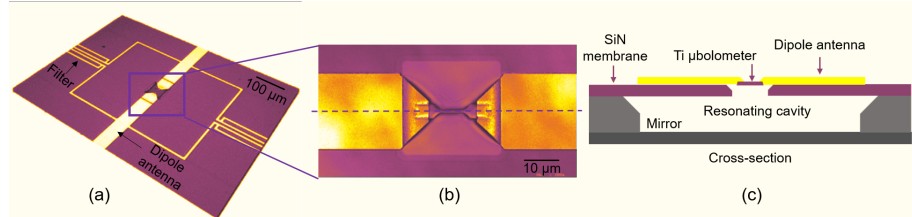

**Figure 1.** The photo of the designed terahertz titanium-based microbolometer with antenna and filters (**a**). The enlarged view of the bridge active part of the microbolometer: the antenna area is approximately 500 μm × 500 μm, the geometry of visible titanium bridge amounts to 12 μm length and 2 μm width (**b**). The cross-section of microbolometric detector. Note that the selectivity was enhanced by adjustment of the dipole antenna geometry and the resonant-cavity design: back side reflection mirror was positioned at the quarter wavelength distance (**c**). Adapted from Refs. [23,24].

Experimental set-ups for terahertz beam profiles imaging and their control are shown in Figure 2. All experiments were performed using raster scanning technique with one pixel detector from linear array of 4 or 32 pixels to obtain beam profile images in *xy* plane. The measuring time per pixel in most experiments was in between 10 ms and 20 ms. The technical details on used equipment and measurement techniques are presented in Section 4.

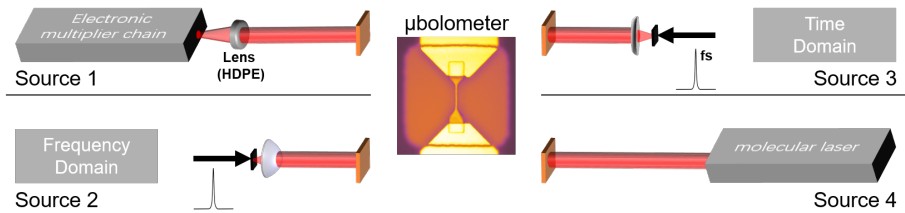

**Figure 2.** Experimental set-up for imaging of terahertz beam profiles using different THz emitters. Source 1—electronic multiplier chain emitting radiation of 0.15 THz, 0.3 THz and 0.6 THz frequencies in continuous wave mode. Source 2—low temperature grown galium arsenide (LTG GaAs) photo-mixer coupled with silicon lens in frequency domain spectrometer delivering radiation from 0.1–2 THz. Source 3—photoconductive antenna on LTG GaAs from time domain spectrometer generating broadband radiation up to 2.5 THz. Source 4—optically pumped continuous wave molecular THz laser emitting the discrete spectrum at 0.76 THz, 1.63 THz and 2.52 THz. Center panel displays photo of Ti-based microbolometer.

## 3. Measurement Results and Discussion

Conventional the diagnostic tool to investigate the THz beam's spatial profile is typical beam imaging, where radiation intensity is scanned in the plane perpendicular to the direction of propagation. To measure the beam profile, focusing optics is usually used to concentrate THz radiation aiming to increase the detected signal. However, any additional focusing element in beam path can distort the beam profile and in many cases cause unwanted interferences. In the THz range, it is of particular importance to use off-axis parabolic mirrors, but then small misalignment can strongly affect the beam profile. As an alternative, one can use plastic lenses; however, they display feature to change focus distance depending on the frequency of incident radiation and they are not so attractive in broadband THz TDS systems, where frequency range, as a rule, extends by two orders of magnitude. To measure the beam profile correctly, one needs to scan unfocused beam that requires high responsivity THz sensor to have sufficient signal-to-noise ratio (SNR). The use of a coherent detector in such a case is limited because the Si lens coupled detector yields limited imaging resolution. Usually, one can employ conventional variable size aperture scanning and Hartmann test techniques to measure far field beam profile [16]. One can note that such approaches are easy to apply and do not require reconfiguration of set-up, nonetheless, reaching high spatial resolution, especially for high THz frequencies, can be rather

complicated because it is determined by the aperture size, and its reduction, consequently, induces the decrease in SNR.

### 3.1. Mode Control in Electronic Multiplier Sources

Initially, the focus of the investigation was attributed to spatial features of THz radiation delivered from electronic multiplier sources. One can note that profile was artificially perturbed by detuning the source from the optimal working regime. The results are shown in Figure 3, where the performance of Ti-mB to record a continuous wave (CW) mode is demonstrated for different polarizations and at three selected frequencies 0.6 THz, 0.3 THz and 0.15 THz. For comparison, measurement results registered by the pyroelectric detector are illustrated in a panel (a) of Figure 3, while the beam profiles at different polarizations are presented in panels (b) and (c). Pyroelectric detector in principle does not detect the polarization of the incident light, thus, registered signal represents total power value of the THz radiation. Despite this fact, the signal values induced in microbolometer and presented in panel (b) are 4–7 times larger than those in panel (a) if compared within the same frequency. In contrast, the microbolometer is polarization-sensitive, thus, not only the mode profiles, but also their fine structure are clearly resolved, which is visible in Figure 3b,c. Comparing the polarization extinction ratio (PER), it amounts to 11 dB, 11.8 dB and 7.8 dB at 0.6 THz, 0.3 THz, 0.15 THz, respectively, which indicates the suitability of the Ti-mB for polarization-sensitive applications. Special emphasis is required to the results obtained at 0.15 THz (panel (c)). Estimates show that the microbolometer connecting wires can act as additional coupling antennas disturbing consequently precise control of the radiation polarization and the mode shape.

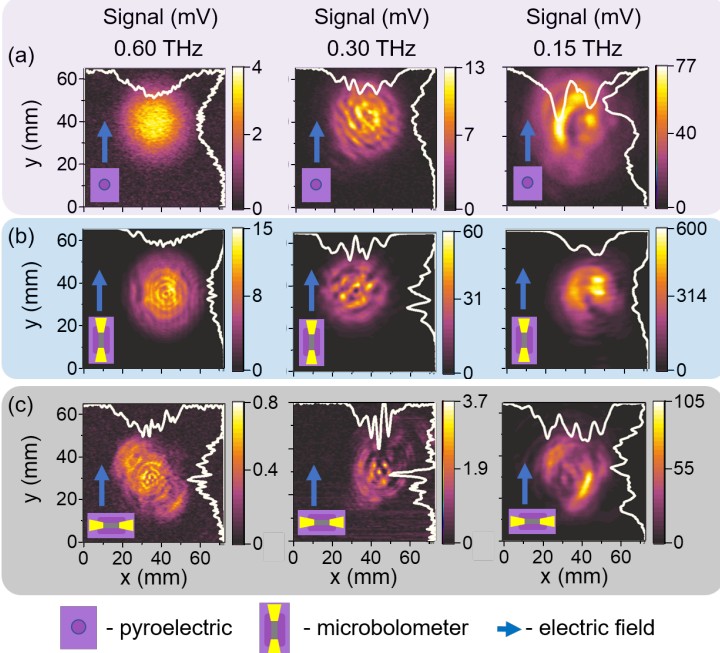

**Figure 3.** Scheme of polarisation orientation and THz beam profiles obtained using electronic multiplier chain source operating in a continuous wave (CW) mode at different frequencies: 0.6 THz, 0.3 THz and 0.15 THz. Reference measurements using pyroelectric detector (**a**); results using microbolometer when the dipole antenna is in parallel to the electric field of the incident light (**b**); results using microbolometer when the dipole antenna is perpendicular to the electric field of the incident light (**c**). White lines indicate intensity cross sections along the relevant axes.

### 3.2. Mode Control in Frequency Domain Spectrometer

One of the important issues to record qualitative spectra, is the precise alignment of the spectroscopic or imaging systems. To illustrate suitability of Ti-mB for such aims, we have performed

a special study using optical mixer-based frequency domain spectrometer. The THz FDS is operating within the frequency range 0.1–2 THz and it delivers radiation of circular polarization of about 1 μW power at 0.2 THz. Therefore, no exceptional attention was attributed to the polarization-related features; however, we restricted ourselves to subTHz range, close to the designed antennas resonances. The mode profile was disturbed intentionally to reveal abilities of the device in recording spatial mode structure. The mode profile was not corrected intentionally. A linear array of the bolometers coupled with antennas exhibiting resonances around 0.2 THz and 0.4 THz was investigated. The results of the measurements are presented in Figure 4. As it is seen, spatial mode profiles are clearly resolved, in particular, at 0.2 THz (panel (a)), where delivered power is close to the maximal value.

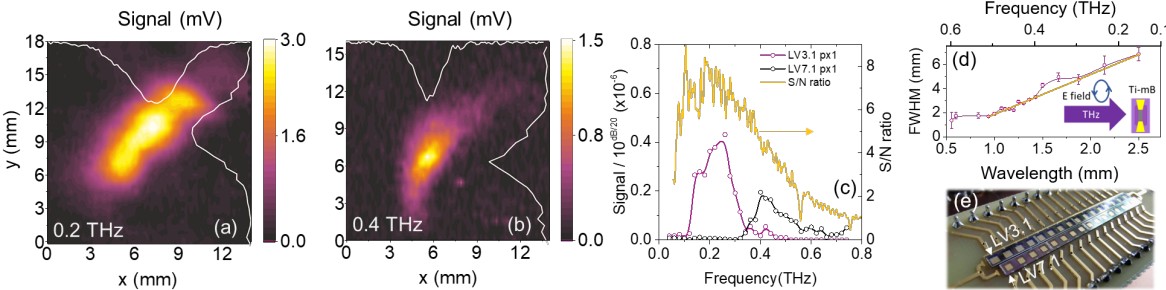

**Figure 4.** THz beam spatial profiles emitted by low-temperature grown GaAs-based emitter in frequency domain spectrometer at 0.2 THz (**a**) and 0.4 THz (**b**) recorded using a linear array of the bolometers coupled with antennas. White lines indicate intensity cross sections along the relevant axes. Spectral characteristics of the detectors with different resonant dipole antennas and spectra normalized to the spectrometer signal-to-noise ratio (SNR) (**c**); full-width half-maximum (FWHM) dependency on the wavelength with linear approximation depicted by the dark yellow line (**d**), configuration of measurement scheme and polarization are depicted in the inset of (**d**). A photo of dual frequency THz linear array, where the top line of pixels is designed for frequencies around 0.2 THz and the bottom line for frequencies around 0.4 THz. The pixel pitch is 2 mm (**e**).

Moreover, the frequency selected exactly falls into relatively broad antenna resonance. The spatial profile presented at 0.4 THz (Figure 4b) is less expressed. It was intentionally done aiming to highlight the convenience and universal operation of the device by detuning the emission frequency to the red side of the spectrum of the antenna resonance as can be evident from panel (c). Despite this fact, the spatial profile can still be precisely controlled although the power absorbed by the microbolometer is significantly reduced. This advantage can further be exploited in determining and monitoring spectral properties of frequency domain spectrometer. It can be illuminated, for instance, via determination of full-width half-maximum (FWHM) dependency on the wavelength of the emission. The results are depicted in Figure 4d. As one can see, the FWHM increases with the wavelength, and the dependence can be approximated by a linear law represented by a dark yellow line (Figure 4d). Thus, Ti-mB microbolometers and their linear array (Figure 4e) can manifest themselves as powerful instruments in fine alignment and control in spectrometers employing weak power CW THz sources.

### 3.3. Mode Control in Time-Domain Spectrometer

In the context of the obtained results in THz frequency-domain, it is reasonable to turn the research into time-domain spectrometry and check a suitability of Ti-mB for this kind of THz systems. In contrast to frequency domain, inherent feature of time-domain spectroscopic systems is their broadband radiation; on the other hand, power of the radiation is in the same range as in THz FDS, i.e., of a few microwatts. Furthermore, THz TDS systems require precise alignment to measure spectra correctly or record images of acceptable quality. In both cases the polarisation of the incident light was parallel to the bolometer axis.

To reveal picture of operation, two types of microbolometers were designed for the experiments: the first one was coupled with resonant antennas for 0.3 THz and the second for 0.7 THz frequency. The choice of the resonant frequencies was motivated by features of the emission—the 0.3 THz line is close to the maximal value in the red side of the spectrum, while 0.7 THz at the blue one, as it can be seen in the measured THz emission spectrum results presented together with experiment schematics in Figure 5a. Sharp lines gracing with the emission spectrum are due to the water vapour absorption in the air. Spatial mode profiles recorded in parallel polarization in respect to microbolometer axis using different coupling antennas are displayed in Figure 5b,c. As it can be seen, the mode profiles can be precisely recorded using both types of antennas. The microbolometer coupled to 0.3 THz resonant antenna removes high-frequency components of the radiation; however, its spatial profile, due to strong low-frequency constituents, reproduces the shape of the mode quite well. The distribution registered at other resonant frequency, around 0.7 THz, shows quite similar picture; however, the spatial mode distribution is narrower in comparison with that at 0.3 THz frequency. One can presume that these data also reproduce correctly the spatial mode distribution despite of the absence of relatively strong low-frequency components recorded. The data obtained allow to infer that the antenna-coupled bolometers can be used for mode profiles control in various optoelectronic THz time-domain spectrometers excited by femtosecond laser pulses [27].

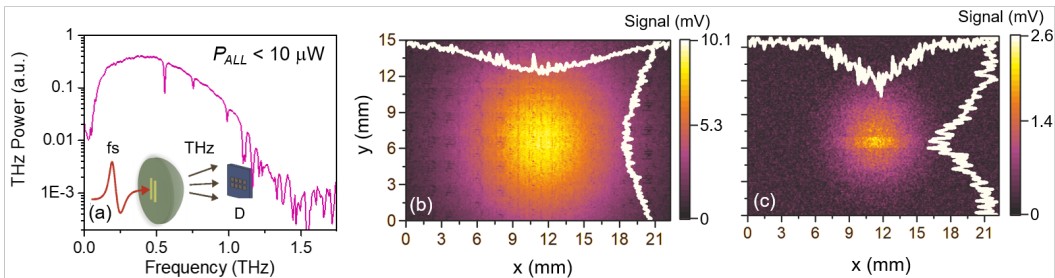

**Figure 5.** Spectrum of THz pulsed emitter (antenna switch) plotted in semi log scale (**a**). The measurement geometry is shown in the inset, where fs denotes to optical femtosecond pulse exciting THz emission from LTG GaAs optoelectronic switch, D labels Ti-mB detector. Note sharp lines caused by water vapour absorption in the air decorating the THz emission spectrum. Beam profiles of THz pulsed emitter obtained with antenna-coupled Ti-mB optimized for resonant frequencies of 0.3 THz (**b**) and 0.7 THz (**c**). White lines indicate intensity cross sections along the relevant axes.

### 3.4. Mode Control in Optically-Pumped Molecular Laser

Since the bolometers exhibit very high sensitivity, it would be of a particular interest to explore their properties far above the coupling antennas resonances. If a mode control could be possible under these unusual conditions, it could extend the device application areas into THz systems with a wider frequency band of operating and hence making the microbolometers more universal. In these experiments, we have chosen THz OPML as a source of radiation delivering discrete spectrum of 0.76 THz, 1.63 THz and 2.52 THz at various power levels. Special attention was given to polarization-related control; it deserves to be mentioned that the spatial mode profile was intentionally disturbed by varying power of the optical pumping and adjusting grating of the pumping laser [28].

The experimental results of bolometers investigation are presented in Figure 6 together with reference data recorded by the commercially available pyroelectric detector. Emission powers at 0.76 THz, 1.63 THz and 2.52 THz frequencies were 1.6 mW, 1.6 mW and 7 mW, respectively, i.e., they were kept in a low level in comparison to conventional powers used in this type of lasers. As it can be seen, reference data presented in Figure 6a can be understood as the averaged signal since the detector used is insensitive to the polarization. The polarization-resolved mode structures—both for parallel and perpendicular polarizations—are displayed in panels (b, c) and allow to analyse in details spatial profiles of the unfocused modes. As the diameter of the laser beam profile amounts to 11 mm, its

spatial deviations can be also nicely distinguished. The estimated polarization extinction ratio was found to be of 2.2 dB, 11.3 dB and 8.9 dB at 0.76 THz, 1.63 THz, 2.52 THz, respectively, suggesting additional application option of the Ti-mB for polarization-sensitive mode profile far above the antenna resonances. Relatively low PER value obtained at 0.76 THz can be explained by two constituents of the mode distinctly observed in Figure 6c.

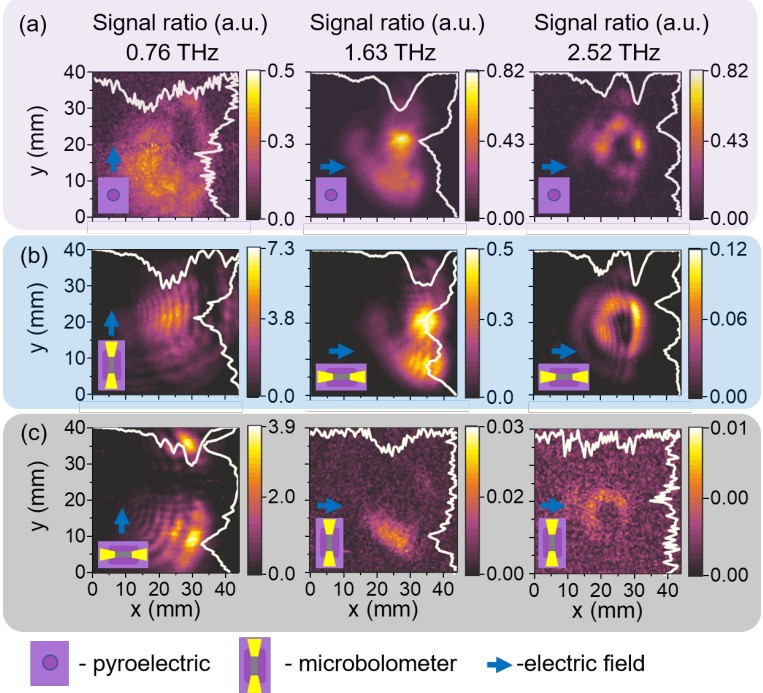

**Figure 6.** THz beam profiles of different polarizations obtained using optically-pumped molecular THz laser operating in a continuous wave mode at frequencies: 0.76 THz, 1.33 THz and 2.52 THz. Reference measurements using pyroelectric detector (**a**); results using microbolometer when the dipole antenna is in parallel to the electric field of the incident light (**b**); results using microbolometer when the dipole antenna is perpendicular to the electric field of the incident light (**c**). White lines indicate intensity cross sections along the relevant axes.

### 3.5. Imaging with Spatial Filtering Methods

To illustrate the sensitive operation of the developed Ti-mBs and their versatile operation using weak power sources we have proceeded in an opposite way that is usually applied in a conventional THz imaging. Here, as a rule, THz images come from scanning the object with the focused beam and registering the focused radiation on the detector making thus induced signal much larger—we have oriented our research for future applications when images without raster scanning, e.g., employing detectors arrays will be required. In such cases, the apertures of imaging optical elements become of a particular importance as the THz wavelength is large in comparison with diameters of optical elements and propagation distances. Therefore, a novel optical solutions [15] are needed allowing to design and manufacture compact optical elements with large aperture sizes and reasonable focal lengths. A special role can be attributed to the case of low absorbing or transparent samples, when the object observation and registration of its internal structure grow into a tremendous challenge [29]. It is related to the fact that power detectors record intensity of the radiation which suffers almost no observable change in the given circumstances, and the shift of the phase of incident radiation becomes a single quantity to be recorded.

In what follows, we employ both single pixel and arrays of the sensitive Ti-mB to record THz images of low absorbing objects under unfocused THz illumination. Both techniques—direct imaging and spatial filtering methods—are used to create a THz image of object inducing phase changes

(uniform amplitude) and that can be registered by intensity sensitive Ti-mB. Conventionally, such filter must be inserted in the Fourier plane in the used imaging system and it will affect only some spatial frequencies of the Fourier spectrum of the object. One can distinguish different methods like bright and dark field [30], Schlieren [31] or phase contrast [32,33] depending on the filter type. They differ in principles and type of the filter; however, all of them permit to create an intensity pattern corresponding to phase changes in the object. The phase contrast method enables linear mapping between phase shift introduced by the object and recorded intensity pattern. In a given study we preferred a phase contrast and dark field methods to register not focused radiation with sensitive Ti-mB.

Results of the investigation together with experimental set-up are given in Figure 7. To elucidate sensing features of the microbolometers, a sample consisting of various low absorbing objects—rubber, paper towel, napkin, sponge, sponge with metal wire inside, plastic (from 1 to 4 layers), aperture of 4 mm in diameter, dots made of silicone, acrylic and wax as well as metal as the reference—was composed and prepared. Measurement set-up for this kind of research is depicted in Figure 7a. As one can see, it is slightly different from that displayed in Figure 2 because of the filter inserted.

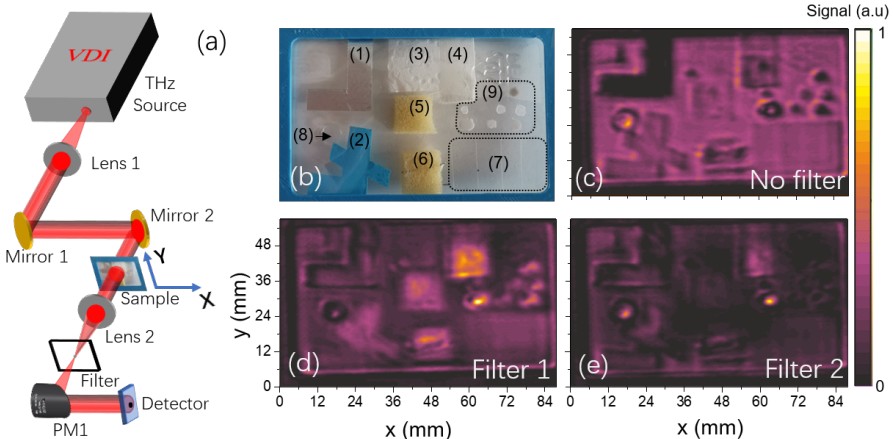

**Figure 7.** Principal scheme of phase contrast imaging set up for 0.3 THz, where L1—HDPE lens with *f* = 12 cm, M1, M2—gold plated flat mirrors, sample, L2—TPX lens with *f* = 6 mm, Filter—(1): ⌀ = 2 mm circle shaped four paper sheets, (2)—⌀ =2 mm circle shaped aluminum foil, PM1—off axis parabolic mirror with *f* = 5 cm (**a**). Photo of the sample with plastic substrate film, it consists of: (1)—metal, (2)—rubber, (3)—paper towel, (4)—napkin, (5)— sponge, (6)—sponge with metal wire inside, (7)—plastic (from 1 to 4 layers), (8)—aperture (diameter: 4 mm), (9)—dots, made of silicone, acrylic and wax (**b**). Direct raster scan image of the sample at 0.3 THz (**c**) . Phase contrast images obtained using filter 1 (**d**) and filter 2 (**e**). THz image pixel size: 0.3 mm × 0.3 mm; images consist of 292 × 190 pixels.

Two types of filters were used in the experiments: the filter 1, consisting of 2 mm circle shaped four paper sheets, was a phase filter introducing a phase retardation of $\pi/2$ only in the central part of the Fourier spectrum and enabling thus to create an image using phase contrast method; second one, filter 2, circle shaped aluminum foil of 2 mm diameter, was an amplitude filter, blocking low spatial frequencies, which resulted in creating an intensity image of phase object having dark background (which is related to removal of low spatial frequencies).

Sample photo and its direct image as well as two corresponding intensity images of phase objects are shown in the corresponding panels (b–d) of Figure 7. It can be easily seen that even direct imaging nicely resolves all the objects illustrating suitability of Ti-mB for imaging of low absorbing objects. As for deeper insight, transparent objects like sponge (5 and 6) and paper tissue (4) are much better visible in case of phase contrast method (filter 1), whilst dark field method (filter 2) gives much better results for imaging of thin rubber glove (2), where we can clearly see edges of the rubber shape.

Finally, it is worth to notice the operation speed of the device. Earlier it was determined that the Ti-mB coupled with a 0.3 THz frequency dipole antenna exhibited the response time of 1 μs range and maintained a sensitivity of 300 V/W and NEP as low as 14 pW/$\sqrt{Hz}$ [26]. If compared with other similar devices based on thermoelectricity-related phenomena, for instance, with uncooled antenna-coupled terahertz detectors based on BiSb/Sb thermocouples with 22 μs response time and NEP of 170 pW/$\sqrt{Hz}$ [34] and bow-tie diodes operating on hot-electrons effects with sensitivity of 5 V/W, NEP of 5.8 nW/$\sqrt{Hz}$ and response time less than 7 ns [35], the described titanium-based microbolometers with readout electronics exhibit high sensitivity exceeding 200 kV/W and relatively short response time (5 microseconds including electronics circuit). Therefore, these parameters together with electronics that can work up to 200 ksps (kilo samples per second) simultaneous sampling speed, makes the Ti-mB promising for real-time fine optical adjustment and mode profile control aims in THz spectroscopic and imaging systems as well as imaging applications and inspection of low-absorbing objects.

## 4. Materials and Methods

Each microbolometer was fabricated from titanium, which was chosen among several appropriate materials, as its micromachining properties were found to be most suitable for silicon processing technology, temperature coefficient and reliability. The sensor was processed on the silicon nitride (SiN) membrane of 2 μm thickness; subsequently, the titanium thermistor film is electro-deposited onto the membrane, along with the aluminum antenna and metal interconnections. The thermistor is suspended in the air by etching out the underlying membrane. The microbolometer was connected either to double-dipole or to log-periodic type THz antenna [23]. Both single pixel bolometers and their linear arrays consisting of 32 pixels were used in the experiments, where pixel size was $2.5 \times 2.5$ mm$^2$, pixel pitch was 2.5 mm. Just one pixel of the linear array was active during all experiments. Sensitivity and NEP of the arrays of sensors exceeds 200 kV/W and less then 20 pW/$\sqrt{Hz}$, respectively. Linear arrays of titanium microbolometers were produced by Luvitera, Ltd, Vilnius, Lithuania.

Electronic multiplier sources (Virginia Diodes, Inc., Charlottesville, Virginia, United States of America) were used to generate radiation of 0.15 THz, 0.3 THz and 0.6 THz frequencies in continuous wave mode with power of 29 mW, 13.2 mW and 0.56 mW, respectively.

Commercially available frequency-domain terahertz spectrometer (Toptica Terascan 780) based on GaAs photo-mixer coupled with silicon lens was engaged in the experiments. Its operation range is within 0.1–2 THz, it delivers radiation of circular polarization of about 1 μW power.

For the experiment a broadband photoconductive low-temperature grown gallium arsenide (LTG-GaAs) antenna coupled with silicon lens from time-domain spectrometer (Teravil, Ltd., Vilnius, Lithuania) was also used. The microbolometer was connected to the lock-in amplifier to modulate signal optically chopped at frequency of 500 kHz. The emitter was pumped with femtosecond Ti-Sapphire optical laser of 100 fs pulse duration and generating broadband THz pulses in frequency range of 0.1–2.5 THz with average emission power of 1.2 μW. The THz beam was investigated by scanning it in the $xy$ plane perpendicular to the direction of propagation with detector being fixed to $xy$ motorized positioning stage. No additional focusing optical components are used in radiation measurements. In beam imaging recording, position synchronized raster scan technique was implemented similar to that described in Ref. [9]. Scanning parameters were selected to enable sufficient SNR with resolution up to $0.1 \times 0.25$ mm$^2$ which is comparable with the wavelength of upper frequency limit of emitted radiation (or the size of the microbolometer).

Optically pumped continuous-wave molecular THz laser FIRL-100 (Edinburg Instruments, Ltd., Livingston, United Kingdom) was used to generate discrete spectrum at 0.76 THz, 1.63 THz and 2.52 THz frequencies with corresponding powers of 1.6 mW, 1.6 mW and 7 mW, respectively. The emitted unfocused radiation was of 11 mm in diameter; the power was artificially reduced varying the power of optical pumping from 20 W down to 9.8 W range.

The absolute power was measured using a calibrated power meter (Thomas Keating Absolute Power Meter System, Version 2, Thomas Keating Ltd., Billingshurst, West Sussex, UK).

Commercially available pyroelectric sensor SPH62 THz with 2 $mm^2$ active area from Spectrum Detector Inc. was used in the experiments.

All experiments were conducted in the ambient air conditions.

## 5. Conclusions

Versatile operation of universal, convenient and easy-to-use antenna coupled sensitive titanium-based microbolometers dedicated to the precise alignment and the control of spatial mode profiles without additional focusing optical components of weak power THz sources are demonstrated. Spatial mode profile control and polarization-resolved mode structures are recorded for different type THz emitters—electronic multiplier sources, optical THz mixers-based frequency domain and femtosecond optoelectronic THz time-domain spectrometers as well as optically pumped molecular THz laser—operating either in continuous wave or pulsed modes. Features of the microbolometers coupled with resonant antennas within 0.15–0.6 THz range are exposed and discussed, and their ability to detect spatial mode profiles far above the antennas resonances, up to 2.54 THz, are explored as well. It is found that a broadband antenna (below 1 THz) coupled microbolometer reproduces well the spatial mode profile of time-domain spectrometer. Polarization-sensitive mode control possibilities are examined in details; suitability of the resonant antenna-coupled bolometers to resolve low-absorbing objects at 0.3 THz is revealed via direct, dark field and phase contrast imaging techniques. The microbolometer, for instance, coupled with a 0.3 THz frequency dipole antenna exhibits the response time of 1 μs range, sensitivity of 300 V/W, and noise-equivalent power of 14 pW/$\sqrt{Hz}$. Microbolometers display corresponding values of 200 kV/W and less then 20 pW/$\sqrt{Hz}$, respectively, and the response time in the 5 μs range. These parameters make the devices promising for real-time fine adjustment of THz spectroscopic and imaging systems as well as imaging applications and inspection of low-absorbing objects.

**Author Contributions:** Conceptualization of the research, G.V. and L.M.; methodology, G.V., L.M. and A.S. (Agnieszka Siemion); software, D.J.; development of antenna-coupled Ti-microbolometers, A.S. (Aleksander Sešek), A.Š., I.K., and J.T.; investigation using electronic sources, L.M., A.S. (Agnieszka Siemion), L.Q. and D.J.; investigation using frequency domain spectrometer, D.S.; investigation using THz time-domain spectrometer, I.K.; investigation using optically-pumped molecular THz laser, D.S., L.Q., and L.M.; data acquisition, L.M. and G.V.; experimental data analysis, L.M., A.S. (Agnieszka Siemion), L.Q., and G.V.; writing—original draft preparation, G.V.; writing—review and editing, L.M., A.S. (Agnieszka Siemion), I.K, and G.V.; visualization, L.M., A.S. (Agnieszka Siemion), L.Q., and D.J.; supervision and coordination, G.V.; project administration, G.V. All authors have read and agreed to the published version of the manuscript.

**Funding:** L.M. and I.K. acknowledge financial support from "KOTERA-PLAZA" project supported from European Regional Development Fund (Grant No. 01.2.2-LMT-K-718-01-0047).

**Acknowledgments:** Authors are very grateful to Rimvydas Venckevičius and Ramūnas Adomavičius for their kind assistance in experiments with femtosecond optoelectronic THz emitters and for illuminating discussions.

**Conflicts of Interest:** The authors declare no conflict of interest.

## Abbreviations

The following abbreviations are used in this manuscript:

| | |
|---|---|
| THz | Terahertz |
| THz ELS | Terahertz electronic multiplier source |
| THz FDS | Terahertz frequency domain spectrometer |
| THz TDS | Terahertz time domain spectrometer |
| THz OPML | Optically pumped molecular THz laser |
| Ti-mB | titanium-based microbolometer |
| CW | continuous wave |
| SNR | Signal-to-noise ratio |
| Ksps | Kilo samples per second |

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
