# Peer review of "Titanium-Based Microbolometers: Control of Spatial Profile of Terahertz Emission in Weak Power Sources"

_applsci, doi:10.3390/app10103400_

Round 1
Reviewer 1 Report
Generally a well written and interesting paper on the use of a fast single pixel micobolometer for beam profiling tested on a wide variety of THz sources.
The following points should be considered in a minor revision:
1) In the introduction and the discussion I miss a more profound review and comparison with beam profiling using 2D camera systems (e.g. micobolometer array, pyrocams…) which is an important alternative to the presented approach.
2) What is the measuring time of the full scans you used for the capture of a beam profile. You only discuss the potential of a high data acquisition in principle but do not give actual numbers.
3) If the measuring time was minutes or hours (what I assume), how did you treat possible intensity variation of the THz source (in particular the FIRL 100)?
4) in Fig 3.-6 you indicate cross sections by white lines. Are those cross section average values or just taken from a specific x, or y position?
Reviewer 2 Report
The paper deals with a scientific important process. An innovative measuring technique for checking the condition is highly desirable.
The paper is well and clearly structured. Only minor revisions are necessary.
Chapter 2: Bolometers’ Design and Experimental Set-ups
To improve the readability, I would recommend to briefly describe the execution of the experiment already here. Because of figure 1 you would otherwise assume that an area detector is used and not a one-pixel detector that is moved in XY-direction.
There is no indication of the measuring time per pixel. Please specify
Chapter 3.1:
As the pyroelectric detector measure the total intensity only, it would be interested to see the total intensity measured by the microbolometer, e. g., by adding the intensities of the two polarizations.
Chapter 3.2:
In figure 4c, you show the spectral characteristics of the two different detectors. It seems that there is a cross talk: please comment this effect for the broad TDS source.
The beam shape of the THz FDS appears very strange. Please compare your results with the results of other papers or a measurement using your pyroelectric detector.
Chapter 3.5:
There is no explanation for the abbreviation “ksps”. Please add it and also check the value.
Chapter 4
Silicon nitride is SiON?
Please specify the linear array: Pixel size and spacing, number of elements, …
Reviewer 3 Report
The authors present a new instrument for controlling spatial profile of THz beam delivered by weak power sources. Ti-based microbolometers have been presented as a THz imaging detector to adjust and control spatial mode profile without any additional focusing optical components. This is pretty innovative, if such an instrument can be directly and easily used in front of the emitter.
At this proposal the authors show different operation modes of Ti-based bolometers, even when coupled with resonant and broadband antennas.
I think that in those sections the results are well discussed and the spatial profile is reported in clear figures. In a couple of cases the data are also compared with data of a pyroelectric detector. I was wondering if it could be possible to improve the section on Time Domain Spectrometer with some similar comparison. Profile-scanning along the direction of distance from the emitter is completely missing. It would be interesting to see the efficiency of the detector as a function of the distance from the emitter.
I think that the most relevant aspect that could be emphasized is the polarization-sensitivity of the presented instrument. Moreover the pure imaging measurements shown in the last section and given by an array of Ti-based bolometers are worthy to be highlighted.
My suggestion could be trying to reorganize the sections in order to give more relevance to the most applicable aspects of your device.
In my opinion, few adjustments could really improve the impact of the manuscript, which is anyhow worthy to be published on Applied Sciences.
